# Trapping of spermine, Kukoamine A, and polyamine toxin blockers in GluK2 kainate receptor channels

Shanti Pal Gangwar [1,6], Maria V. Yelshanskaya[1,6], Muhammed Aktolun[2,6], Laura Y. Yen[1,3], Thomas P. Newton [1,4], Kristian Strømgaard [5], Maria G. Kurnikova [2] & Alexander I. Sobolevsky[1]

Kainate receptors (KARs) are a subtype of ionotropic glutamate receptor (iGluR) channels, a superfamily of ligand-gated ion channels which mediate the majority of excitatory neurotransmission in the central nervous system. KARs modulate neuronal circuits and plasticity during development and are implicated in neurological disorders, including epilepsy, depression, schizophrenia, anxiety, and autism. Calcium-permeable KARs undergo ion channel block, but the therapeutic potential of channel blockers remains underdeveloped, mainly due to limited structural knowledge. Here, we present closed-state structures of GluK2 KAR homotetramers in complex with ion channel blockers NpTx-8, PhTx-74, Kukoamine A, and spermine. We find that blockers reside inside the GluK2 ion channel pore, intracellular to the closed M3 helix bundle-crossing gate, with their hydrophobic heads filling the central cavity and positively charged polyamine tails spanning the selectivity filter. Molecular dynamics (MD) simulations of our structures illuminate interactions responsible for different affinity and binding poses of the blockers. Our structures elucidate the trapping mechanism of KAR channel block and provide a template for designing new blockers that can selectively target calcium-permeable KARs in neuropathologies.

Kainate receptors (KARs) are tetrameric ion channels that belong to the family of ionotropic glutamate receptors (iGluRs)[1]. Functional KARs are assembled from subunits GluK1-5, encoded by *GRIK1-5* genes[2–4]. GluK1-3 are considered 'primary' subunits as they can form functional homomeric and heteromeric channels. On the other hand, the 'secondary' GluK4-5 subunits only form functional heterotetramers with GluK1-3 subunits[5]. KARs are expressed ubiquitously throughout the central nervous system (CNS) and are distributed both pre- and post-synaptically, playing a crucial role in regulating neuronal activity[6]. Consequently, activation of KARs elicits a diverse array of pre- and post-synaptic effects on glutamatergic and gamma-aminobutyric acid (GABA)ergic synaptic transmission[7,8]. Whereas presynaptic KARs modulate neurotransmitter release and postsynaptic KARs mediate excitatory neurotransmission, extrasynaptic KARs are involved in controlling neuronal excitability[2,9–13]. The functional diversity of KARs, including ion selectivity, ligand specificity, and kinetics, is further

[1]Department of Biochemistry and Molecular Biophysics, Columbia University, 650 West 168th Street, New York, NY 10032, USA. [2]Department of Chemistry, Carnegie Mellon University, Pittsburgh, PA 15213, USA. [3]Cellular and Molecular Physiology and Biophysics Graduate Program, Columbia University Irving Medical Center, 630 West 168th Street, New York, NY 10032, USA. [4]Integrated Program in Cellular, Molecular and Biomedical Studies, Columbia University Irving Medical Center, 630 West 168th Street, New York, NY 10032, USA. [5]Center for Biopharmaceuticals, Department of Drug Design and Pharmacology, University of Copenhagen, Jagtvej 162, DK-2100 Copenhagen, Denmark. [6]These authors contributed equally: Shanti Pal Gangwar, Maria V. Yelshanskaya, Muhammed Aktolun. ✉e-mail: as4005@cumc.columbia.edu

expanded through RNA editing, alternative splicing of subunit transcripts, and interactions with auxiliary subunits[14–19]. Thus far, KARs have been found to associate with Neto-type auxiliary subunits, Neto1-2, which increase current amplitude, alter agonist efficacy, slow the deactivation and desensitization rates, increase the channel open probability, and modulate neuronal localization[10,20–23].

Previous studies demonstrated that perturbations in KAR function are implicated in neurological and neurodegenerative disorders, including epilepsy, ischemia, stress, anxiety, intellectual disability, and pain[2,24–29]. Small molecule inhibitors of KARs might therefore have therapeutic potential. One of the prospective classes of KAR inhibitors are ion channel blockers that act on unedited receptors which have glutamine instead of arginine at the functionally important Q/R-site that determines calcium permeability and plays a critical role in glutamate-induced neurotoxicity[30,31]. Two types of KAR channel blockers have been identified: polyamines and polyamine-containing toxins or toxin-like molecules[32].

Polyamines are ubiquitous in bacterial, plant, and animal cells and often serve as blockers of cation-selective ion channels[33–35]. Polyamine block of ion channels, primarily by endogenous spermine (SPM) and spermidine, was first discovered for potassium channels[36] but subsequently revealed for iGluRs[37], nicotinic acetylcholine receptors[38], and cyclic nucleotide-gated (CNG) ion channels[39]. Due to their cationic nature, polyamines bind to negatively charged regions of biomolecules, including electronegative pores of cation-selective voltage- and ligand-gated ion channels, often with micromolar affinity[35,38–43]. At glutamatergic synapses of the developing and mature CNS, polyamines act as permeant ion channel blockers of both α-amino-3-hydroxy-5-methyl-4-isoxazolepropionic acid receptors (AMPARs), another member of the iGluR family, and KARs[35,44,45]. Accordingly, cytoplasmic polyamines have been recognized as important determinants of neuronal signaling that regulate action potential firing rates[46] as well as the strength of neurotransmission[47,48].

The second category of ion channel blockers that act at both AMPARs and KARs includes extracellularly-applied exogenous polyamine- or acylpolyamine-containing toxins, such as argiopin, also known as argiotoxin-636 (AgTx-636)[49–51], from the spider *Argiope lobata*, Joro spider toxin JSTX-3[52,53] from *Nephilia clavata*, philanthotoxin-433 (PhTx-433) from the wasp *Philanthus triangulum*[54,55], and *Nephila* spider toxin-8 (NpTx-8)[51]. Neuroprotective properties demonstrated by some of these toxins[56–58] inspired the development of synthetic analogs, including PhTx-433 derivatives[56,59–64] like PhTx-343, PhTx-56, and PhTx-74, the Joro spider toxin analog 1-naphthyl acetyl spermine (NASPM)[65–67], which reduces oxidative stress and protects neurons after ischemia[68,69], and IEM-1460 adamantane derivative[67,70] that attenuates epileptic seizures[71,72] and reduces visceral pain[73,74] and receptor activity related to schizophrenia[75].

Previously, the molecular mechanism of ion channel block was characterized structurally for exogenous AMPAR channel blockers AgTx-636, IEM-1460, and NASPM[67]. More recently, synaptic complexes of GluA2 homotetramer with auxiliary subunits γ5 and CNIH2 were shown to be blocked by the endogenous polyamine spermidine[76]. However, the structural basis of KAR channel block by polyamines and polyamine-related molecules has remained elusive.

Here, we embark on studies of KAR channel block using single-particle cryo-EM, electrophysiology, and MD simulations. We solve cryo-EM structures of homotetrameric GluK2 KAR in complex with PhTx-74, NpTx-8, SPM, and bioactive compound Kukoamine A (KukoA) (Fig. 1a), a spermine alkaloid originally isolated from *Lycium chinense* (*goji berry*), which is known for its hypotensive effect[77]. It has diverse biological activities, including anticancer, neuroprotective, and anti-inflammatory properties[78,79]. We show that PhTx-74, NpTx-8, SPM, and KukoA act as trapping blockers. We also perform MD simulations of our structures in near physiological conditions, including room temperature and the presence of electrolytes and lipid bilayers, to determine specific interactions of blocker molecules with KARs and to investigate the blocker pose stability and binding site variability. In addition, we compute the pore dimensions and electrostatic profiles for the KAR-blocker complexes to decipher the molecular determinants of blocker binding in the ion channel pore.

## Results

### Functional characterization of KAR channel block

We used whole-cell patch-clamp recordings to study the function of calcium-permeable homotetrameric GluK2 receptors transiently expressed in HEK 293 cells. In the presence of the positive allosteric modulator Concanavalin A (ConA), which dramatically reduces KAR desensitization, the agonist kainic acid (KA) elicited large inward currents with a steady-state value of $I_{Control}$ (Fig. 1b, black traces). In the presence of the ion channel blockers NpTx-8, PhTx-74, KukoA, or SPM, the current amplitude was reduced and characterized by a smaller steady-state value, $I_{Block}$ (Fig. 1b, red traces).

NpTx-8, PhTx-74, KukoA, and SPM inhibited GluK2-mediated currents in a concentration-dependent manner, with stronger inhibition observed at higher blocker concentrations (Fig. 1c). The potency of the blockers varied widely, with the half-maximal inhibitory concentration, $IC_{50} = 0.51 \pm 0.01\,\mu M$ ($n = 9$) for NpTx-8, $IC_{50} = 7.53 \pm 0.35\,\mu M$ ($n = 10$) for PhTx-74, $IC_{50} = 704 \pm 31\,\mu M$ ($n = 7$) for KukoA, and $IC_{50} = 7.77 \pm 0.51\,mM$ ($n = 8$) for SPM. Such dramatic differences in blocker potency are typically associated with faster dissociation kinetics for low-affinity blockers and slower ones for high-affinity blockers. When the blocker dissociation kinetics are faster than channel closure, simultaneous termination of the blocker and agonist application results in the appearance of resurgent, or "hook", currents carried by channels that have been relieved from the block but are still awaiting pore closure preceding agonist dissociation[80–82]. Accordingly, the highest amplitude hook currents were observed for the lowest-affinity blocker SPM. In contrast, no hook currents were observed for the highest-affinity blocker NpTx-8 (Fig. 1b). The intermediate affinity blockers, PhTx-74 and KukoA, showed hook currents of intermediate amplitude (Fig. 1b).

While the presence of the hook currents is strongly dependent on the blocker dissociation rate, their amplitude and kinetics are also determined by the interaction of the blocker with channel gating[80–83]. If a blocker prevents channel closure or desensitization, the hook current may have a higher amplitude and decay much slower than the control tail current. On the contrary, for a blocker that does not interfere with channel gating (trapping blocker), the hook current amplitude does not exceed the amplitude of the control tail current, and the current decay kinetics are similar. Therefore, the hook currents observed for PhTx-74, KukoA, and SPM belong to the second category, indicating that these molecules represent trapping blockers of the GluK2 channels.

Inhibition of GluK2-mediated currents by the positively charged molecules of NpTx-8, PhTx-74, KukoA, and SPM was also voltage-dependent, consistent with the mechanism of channel block. In the absence of blockers, the GluK2-mediated currents showed strong inward rectification due to the block by intracellular polyamines (Fig. 1d). In the presence of NpTx-8, PhTx-74, KukoA, or SPM, the current amplitude was reduced but the extent of current reduction, represented by $I_{Block} / I_{Control}$, was much stronger at negative compared to positive membrane voltages, reflected in the values of the current rectification index, $(I_{Block} / I_{Control})_{-60mV} / (I_{Block} / I_{Control})_{+60mV}$, being much smaller than 1 (Fig. 1e). The similar values of the current rectification index for NpTx-8, PhTx-74, KukoA, and SPM suggest that all four blockers likely bind to approximately the same or overlapping binding sites inside the GluK2 ion channel pore.

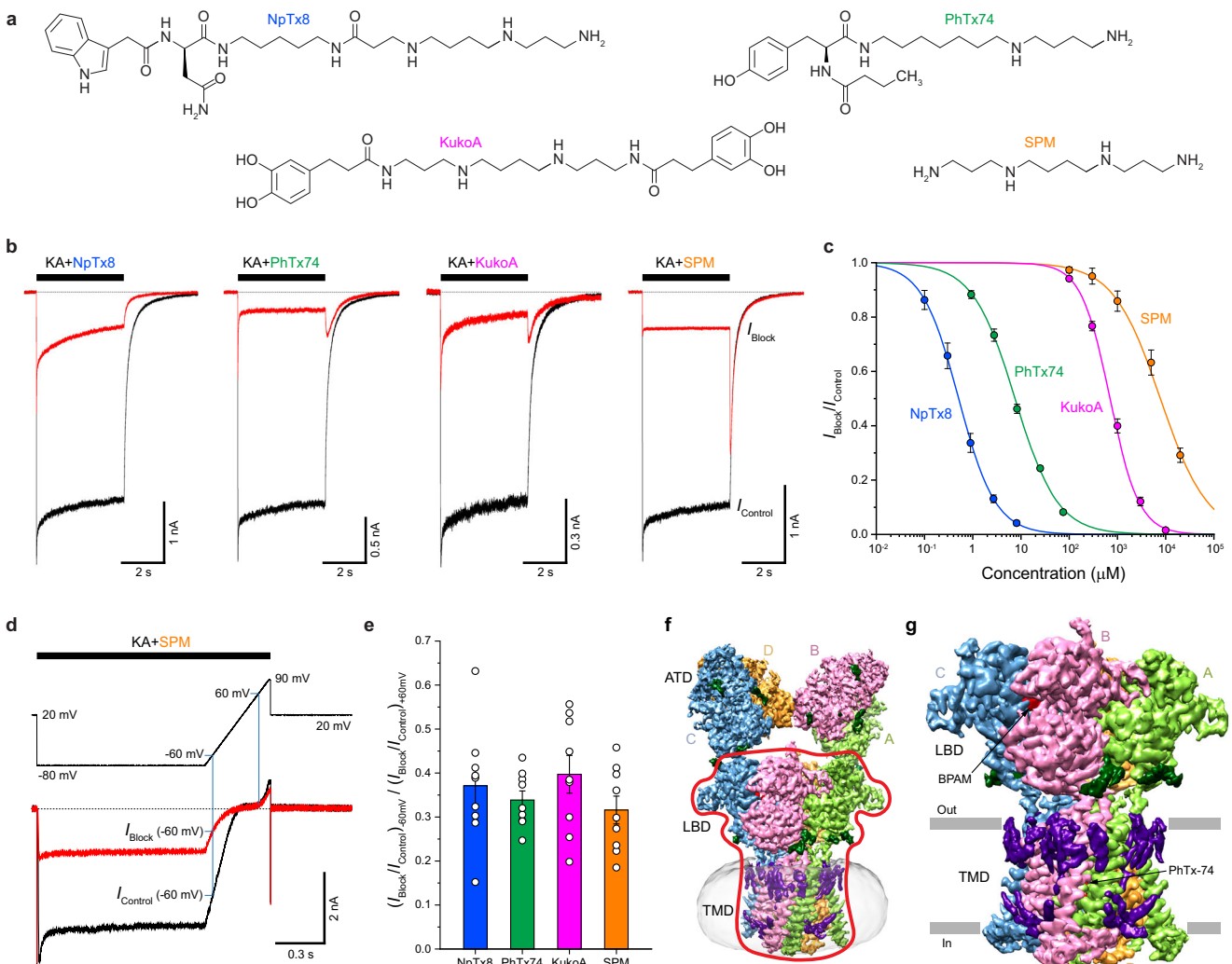

**Fig. 1 | Functional characterization of GluK2 channel block by NpTx-8, PhTx-74, kukoamine A and spermine and cryo-EM. a** Chemical structures of NpTx-8, PhTx-74, KukoA, and SPM. **b** Whole-cell patch-clamp currents recorded at −60 mV membrane potential from HEK293 cells expressing GluK2 in response to application of KA alone (black traces) or with 2.7 μM NpTx-8, 75 μM PhTx-74, 3 mM KukoA or 20 mM of SPM (red traces). **c** Concentration-dependencies of GluK2 block by NpTx-8 (blue), PhTx-74 (pink), KukoA (green) and SPM (orange) fitted to the logistic equation with the parameters: $IC_{50} = 0.51 \pm 0.01$ μM and $n_{Hill} = 1.14 \pm 0.01$ ($n = 9$) for NpTx-8, $IC_{50} = 7.53 \pm 0.35$ μM and $n_{Hill} = 0.99 \pm 0.04$ ($n = 10$) for PhTx-74, $IC_{50} = 704 \pm 31$ μM and $n_{Hill} = 1.45 \pm 0.05$ ($n = 7$) for KukoA, and $IC_{50} = 7.77 \pm 0.51$ mM and $n_{Hill} = 0.89 \pm 0.04$ ($n = 8$) for SPM. Data are mean ± SEM. Source data are provided. **d** Voltage-dependence of the whole-cell current recorded from an HEK293

cell expressing GluK2 in response to a step of −80 mV and then a −80 to +90 mV voltage ramp during application of KA alone (black trace) or with 20 mM SPM (red trace). Vertical blue lines indicate −60 mV and +60 mV voltages, at which rectification index measurements were taken. **e** Rectification index, $(I_{Block} / I_{Control})_{-60mV} / (I_{Block} / I_{Control})_{+60mV}$, calculated at 1.5 μM NpTx-8, 30 μM PhTx-74, 2 mM KukoA and 20 mM of SPM. Data are mean ± SEM. The number of biologically independent measurements, $n = 9$ for NpTx-8, $n = 9$ for PhTx-74, $n = 9$ for KukoA, and $n = 9$ for SPM. Data are representative of two independent experiments. **f** Cryo-EM density for full-length GluK2$_{PhTx74}$, with GluK2 subunits colored green, pink, blue and orange. The red contour envelopes the LBD-TMD region. **g** Cryo-EM density for the LBD-TMD region with the micelle and ATD signals masked out (see the red contour in **f**).

## Cryo-EM reconstruction of GluK2 in the presence of ion channel blockers

To uncover the molecular details of NpTx-8, PhTx-74, KukoA, and SPM binding inside the ion channel pore, we subjected GluK2 to single-particle cryo-EM in the presence of each ion channel blocker individually and positive allosteric modulator BPAM344 (BPAM), which stabilizes the upper D1-D1 ligand-binding domain (LBD) dimer interface (Supplementary Figs. 1–5, Table 1). A typical 3D reconstruction of the full-length GluK2 showed 3-layer architecture, characteristic of AMPARs and KARs[3,84–89], with the amino terminal domain (ATD) at the top, LBD in the middle, and the transmembrane domain (TMD) at the bottom (Fig. 1f). The ATD and LBD layers, connected by flexible linkers, showed a high degree of relative movement, signified by blurriness of the ATD layer compared to the LBD-TMD in 2D class averages (Supplementary Figs. 1–4). To improve the quality of 3D reconstructions,

we therefore performed classification and refinement by focusing on the LBD-TMD region only (Fig. 1f, g). Indeed, cryo-EM reconstruction of the LBD-TMD region alone resulted in improved map quality, with well-resolved densities for BPAM, carbohydrates, and annular lipids (Fig. 1g; Supplementary Fig. 5a–c).

Overall, the 2.81–3.75 Å reconstructions of GluK2 in the presence of BPAM and ion channel blockers were similar to one another and to the reconstruction of GluK2 in the presence of BPAM only, which yielded the closed-state structure, GluK2$_{closed}$ (PDB ID: 8FWS), published before[86]. The biggest difference between the reconstructions was the presence of variably shaped densities inside the pore of the TMD (Fig. 2). While no density was observed in GluK2$_{closed}$, reconstructions for the samples made in the presence of the blockers revealed densities in the middle of the pore that were roughly matching the chemical structure of the corresponding blocker (Fig. 1a;

**Table 1 | Cryo-EM data collection, refinement, and validation statistics**

| Structure | GluK2$_{NpTx8}$ | GluK2$_{PhTx74}$ | GluK2$_{KukoA}$ | GluK2$_{SPM}$ |
|---|---|---|---|---|
| EMDB accession code | EMD-47296 | EMD-47295 | EMD-47298 | EMD-47297 |
| PDB accession code | 9DXR | 9DXQ | 9DXT | 9DXS |
| Data collection and processing | | | | |
| Voltage (kV) | 300 | 300 | 300 | 300 |
| Electron exposure (e$^-$Å$^{-2}$) | 50 | 40 | 47.25 | 54.4 |
| Reported pixel size (Å) | 1.055 | 0.832 | 1.069 | 0.835 |
| Processing software | CryoSPARC v4 | CryoSPARC v4 | CryoSPARC v4 | CryoSPARC v4 |
| Symmetry imposed | C2 | C2 | C2 | C2 |
| Final particle images (no.) | 199,303 | 238,611 | 211,509 | 193,200 |
| Map resolution (Å) | 3.10 | 2.81 | 3.75 | 3.55 |
| FSC threshold | 0.143 | 0.143 | 0.143 | 0.43 |
| Refinement | | | | |
| Initial models used (PDB code) | 8FWS | 8FWS | 8FWS | 8FWS |
| Model resolution (Å) | 3.10 | 2.81 | 3.75 | 3.55 |
| Mask CC | 0.79 | 0.83 | 0.78 | 0.84 |
| Volume CC | 0.72 | 0.81 | 0.78 | 0.83 |
| Map sharpening B factor (Å$^2$) | −107.1 | −133.4 | −150.8 | −146.5 |
| B factors (Å$^2$) | | | | |
| Protein | 102.77 | 99.39 | 123.85 | 129.08 |
| Ligands | 79.77 | 77.55 | 127.62 | 140.84 |
| R.m.s. deviations | | | | |
| Bond lengths (Å) | 0.010 | 0.011 | 0.010 | 0.010 |
| Bond angles (°) | 1.477 | 1.423 | 1.494 | 1.418 |
| Model composition | | | | |
| Non-hydrogen atoms | 15333 | 15555 | 15138 | 15294 |
| Protein residues | 1784 | 1784 | 1784 | 1784 |
| Ligands | | | | |
| 2J9 (BPAM344) | 4 | 4 | 4 | 4 |
| BMA | 4 | 4 | 4 | 4 |
| NAG | 16 | 16 | 16 | 14 |
| POV | 16 | 16 | 8 | 12 |
| CL | – | 2 | – | – |
| NA | – | 6 | – | – |
| CLR | – | 8 | 8 | 8 |
| Channel blocker | 1 (NTX) | 1 (PTX) | 1 (KUK) | 1 (SPM) |
| Validation | | | | |
| MolProbity score | 1.87 | 1.75 | 1.97 | 1.83 |
| Clash score, all atoms | 4.67 | 4.14 | 4.54 | 4.38 |
| Outliers rotamers (%) | 1.16 | 1.03 | 0.26 | 0.90 |
| Ramachandran plot | | | | |
| Favored (%) | 88.66 | 90.29 | 79.28 | 87.50 |
| Allowed (%) | 10.78 | 9.37 | 17.57 | 11.60 |
| Outliers (%) | 0.56 | 0.34 | 3.14 | 0.90 |

Supplementary Fig. 5d–f. Each density had an approximate 4-fold rotational symmetry that characterizes the channel of kainate receptor at this location and can fit the blockers in four equivalent poses, different by 90-degree rotation around the axis of local symmetry (Supplementary Fig. 5e, f). Unfortunately, symmetry expansion as well as focused classification and refinement techniques did not improve these densities, most probably because of the very small size of blocker molecules. The heterogeneity of blocker binding poses due to the local 4-fold rotational symmetry as well as the dynamic nature of binding (see MD simulations section below) are the likely reasons why the blocker densities are somewhat weaker than the density for the surrounding protein.

Reconstruction in the presence of SPM showed a narrow density matching the short and thin chemical structure of this polyamine. For KukoA, the density had a dumbbell shape, consistent with the two-headed structure of this molecule. For NpTx-8 and PhTx-74, the extracellular end of the density had a bulky cap, which likely represents averaged conformations of the blockers hydrophobic head, while its intracellular region included four prongs, likely representing four different orientations of the positively charged polyamine tail (Fig. 2). No other densities apparent in the presence but not in the absence of ion channel blockers were found in our cryo-EM reconstructions, strongly suggesting that NpTx-8, PhTx-74, KukoA, and SPM binding occurs in the ion channel pore only.

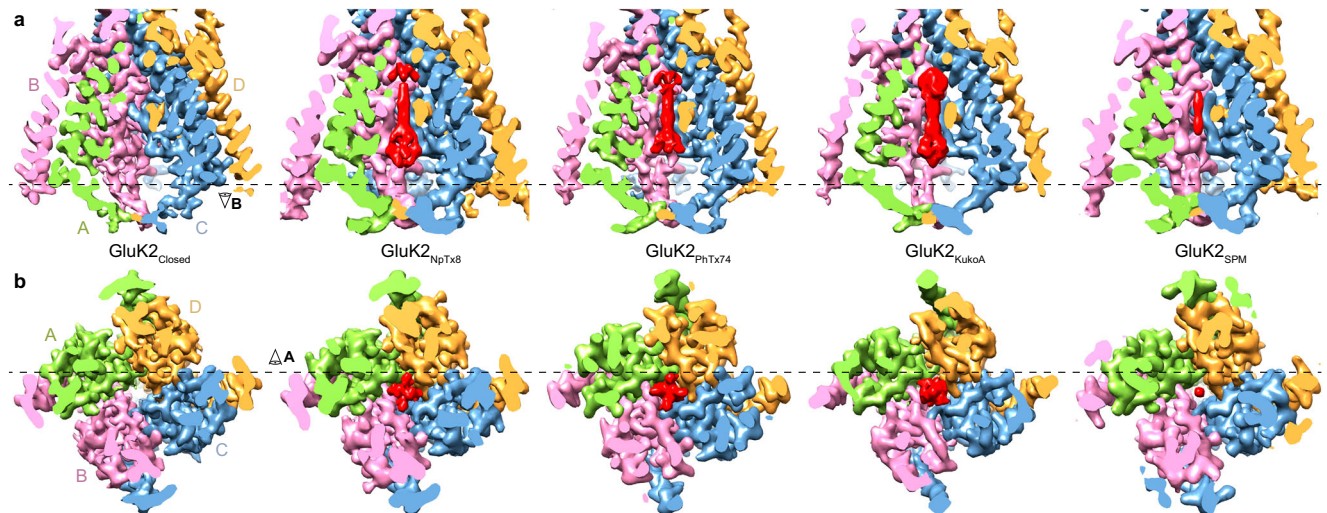

**Fig. 2 | TMD density in the absence and presence of ion channel blockers.**
Coronal (**a**) and transverse (**b**) plane views of cryo-EM density for the TMD in
GluK2$_{closed}$, GluK2$_{NpTx8}$, GluK2$_{PhTx74}$, GluK2$_{KukoA}$, and GluK2$_{SPM}$, with GluK2
subunits colored green, pink, blue and orange, and blockers red. The dashed black
line in **a** shows where the density is cut away in **b**, while the dashed black line in
**b** shows where the density is cut away in **a**.

## Structures of GluK2 and blocker binding site

We built atomic models of the GluK2 LBD-TMD region for the cryo-EM
reconstructions in the presence of the different ion channel blockers.
Guided by the cryo-EM density (Supplementary Fig. 5), we built LBDs,
LBD-TMD linkers, and TMDs, including the intracellular cap and the
C-terminal helix, often absent in KAR reconstructions, as well as lipids,
carbohydrates, and ion channel blockers in the middle of the ion
channel pore (Fig. 3a–c). Each LBD exhibits a bilobed architecture
comprising the upper (D1) and lower (D2) lobes, revealed by crystal
structures of the isolated LBD[90,91] and by cryo-EM structures of full-
length KARs[3,85–89]. The LBD layer has a dimer of A/D and B/C dimers
arrangement, observed previously in AMPAR structures[84,92,93], and
structures of KARs in the closed[3,85,86,88] and open[94] states.

Four TMDs form a cation-selective ion channel, composed of the
M1, M3, and M4 transmembrane helices and a re-entrant M2-loop
between M1 and M3 (Fig. 3c, d). M1 and M4 form the ion channel
periphery, while the extended region of M2 and the C-terminal half of
M3 line the ion channel pore. All four ion channel blockers fit the
section of the ion channel pore intracellular to the M3 helices bundle
crossing, which includes the central cavity lined by side chains of L645,
I648, S649, and T652, and the selectivity filter contributed by side
chains Q621 and E625 and backbone carbonyls of Q621, Q622, G623,
S624, and E625 (Figs. 3d and 4a–d). The bulky hydrophobic heads of
NpTx-8 and PhTx-74 occupy the central cavity, while their polyamine
tails stretch nearly the entire length of the selectivity filter (Figs. 3d and
4a, b). Based on the four-pronged cryo-EM densities corresponding to
NpTx-8 and PhTx-74 (Fig. 2), the polyamine tails of these blockers do
not align with the pore axis but instead lean towards the selectivity
filter walls adopting one of four nearly identical positions in the
pseudo-4-fold symmetrical ion channel. KukoA has two identical bulky
heads at the polyamine ends that each have equal possibility to fit the
central cavity, while the other interacts with E625 at the intracellular
pore entrance (Fig. 4c). Two upper thirds of the SPM molecule fit the
extracellular part of the selectivity filter, with the corresponding den-
sity aligned with the pore axis (Fig. 2). The density is much weaker for
the lower third of SPM, which likely adopts numerous conformations
in the intracellular region of the selectivity filter (Fig. 4d).

When superposed with the closed state structure GluK2$_{closed}$
(Fig. 4e), all four blocker-bound structures show nearly identical con-
formation of the channel pore (Fig. 4f, Supplementary Fig. 6), indi-
cating that the structures were solved in the blocker-bound closed

state. Notably, when the open-state structure GluK2$_{open}$ (PDB ID: 9B35;
Fig. 4g) is superposed with the blocker-bound structures (Fig. 4h), the
blocker binding region appears the same as well because the differ-
ences associated with channel opening start extracellularly to the
blocker binding site with M3 helices bending at the gating hinge. To
further validate our models of blocker binding in the pore region and
identify the critical interactions between the blockers and the channel,
we performed MD simulations.

## Molecular dynamics simulations and blocker-channel interactions

The cryo-EM structures GluK2$_{NpTx8}$, GluK2$_{PhTx74}$, GluK2$_{KukoA}$, and
GluK2$_{SPM}$ were used as initial models for MD simulations. For each
structure, we built a simulated system (Fig. 5) and used MD protocols
similar to those used recently for simulating the GluK2$_{closed}$ and
GluK2$_{open}$ structures[86,94] (see Methods, Supplementary Tables 1–3). All
systems were stable throughout the entirety of the MD simulations
(Supplementary Fig. 7). Short equilibrium simulations were designed
to refine the blocker binding poses and explore the dynamics of these
small molecules inside the ion channel pore. Some mobility of the
blocker molecules during MD simulations was expected due to a crude
and averaged appearance of the cryo-EM density for the blocker
molecules (Fig. 2). While all blocker molecules remained in the region
determined by the corresponding structures, they exhibited different
dynamics and varying extents of binding pose alteration (Fig. 6a).

In general, all blockers formed several stable hydrogen bonds in
the narrow region of the selectivity filter via the nitrogen atoms of their
polyamine tails, arranged in a pattern similar to that observed for SPM.
In addition, the hydrophobic bulky groups of NpTx-8, PhTx-74, and
KukoA formed several stable hydrogen bonds and hydrophobic con-
tacts in the channel central cavity. Specifically, NpTx-8 formed stable
hydrogen bonds in the selectivity filter with the backbone carbonyl of
S624, the side chain of Q622, and the carbonyl oxygen of Q621 in an
asymmetric fashion. NpTx-8 also retained a hydrogen bond with the
E625 side chain at the intracellular pore entrance. Its headgroup
formed a stable hydrophobic contact with T652, which is located at the
extracellular boundary of the central cavity, as well as hydrogen bonds
and hydrophobic contacts with L645, I648, and S649 of one subunit,
and simultaneously, with the identical residues of the diagonally
opposing subunit (Fig. 6a). Contacts of the NpTx-8 indole group with
the protein excluded water from the central cavity and thus enhanced

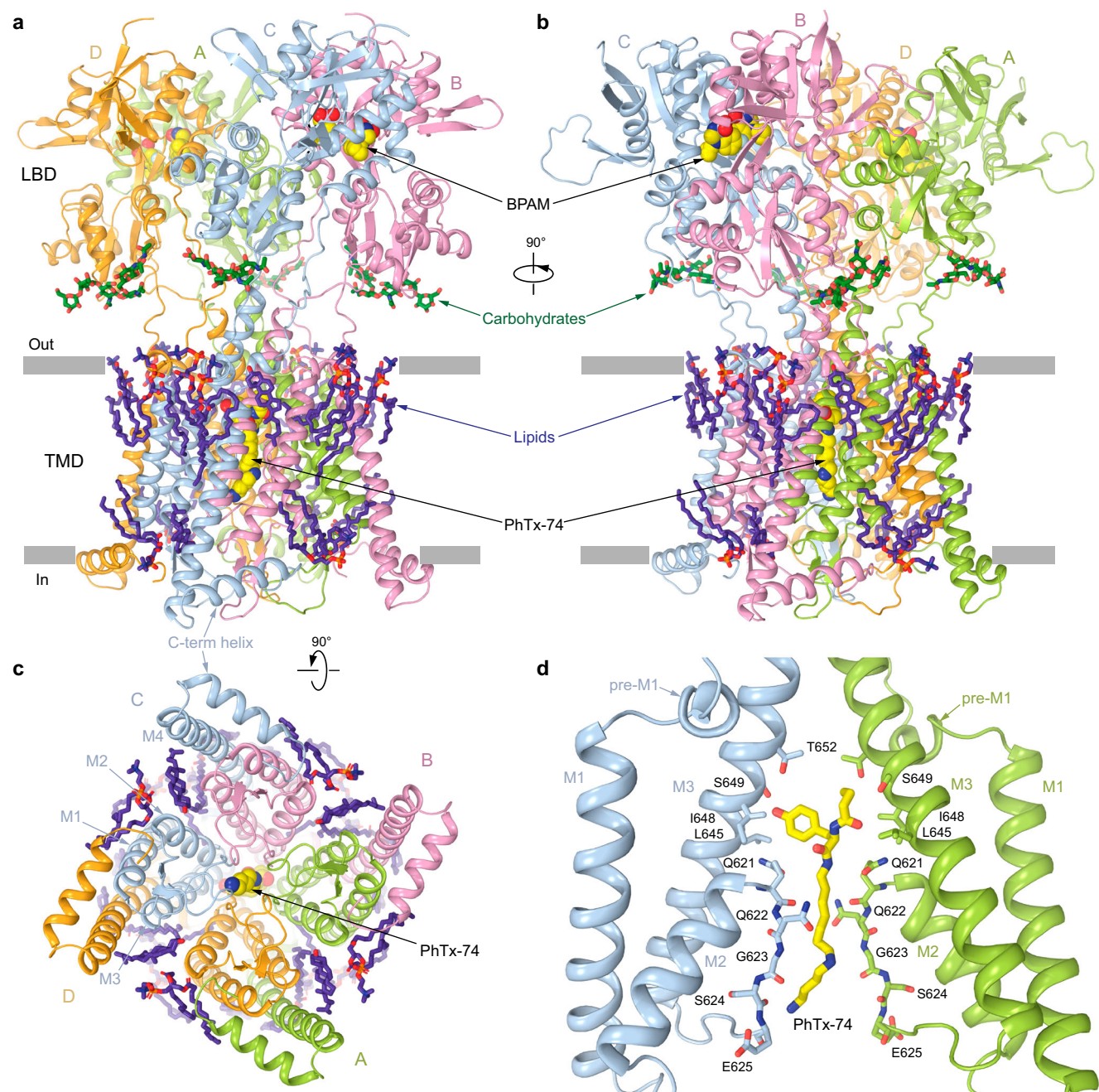

**Fig. 3 | GluK2$_{PhTx74}$ structure and PhTx-74 binding site in the channel pore.**
GluK2$_{PhTx74}$ structure viewed parallel to the membrane (**a**, **b**) and intracellularly (**c**).
The molecules of BPAM and PhTx-74 are shown as space-filling models (yellow),
carbohydrates (green) and lipids (purple) in sticks. **d** Closeup view of the PhTx-74
binding site in the channel pore. Only two (A and C) of four subunits are shown,
with the front and back subunits (B and D) removed for clarity. The molecule of
PhTx-74 (yellow) and residues involved in its binding (blue and green) are shown in
sticks.

the hydrophobic effect (Fig. 6b). The pose of NpTx-8 was asymmetric,
favoring the entropy of binding to the highly degenerate binding site in
the ~4-fold symmetrical homotetrameric ion channel. In longer simu-
lations, the NpTx-8 pose will likely switch between similar residues in
other subunits, which is entropically favorable without compromising
binding. Among the four simulated structures, NpTx-8 formed the
most extensive network of contacts with GluK2, corroborating its
highest potency.

Compared to the starting pose in the cryo-EM model, PhTx-74
dramatically changed its pose during the MD simulations to maximize
its interactions with the channel pore and formed several hydrogen
bonds and hydrophobic interactions with the residues lining the

central cavity (Fig. 6a). The resulting pose formed contacts across the
channel pore. At the same time, water molecules were excluded from
the central cavity (Fig. 6b). The tail of the blocker formed hydrogen
bonds with the backbone carbonyls of Q621 and Q622, which were not
present in the cryo-EM structural model. Overall, PhTx-74 formed
fewer interactions with the channel pore than NpTx-8, explaining its
somewhat weaker potency and stronger discrepancy with the cryo-EM
model. This example of discrepancy between cryo-EM and MD results
illustrates limitations of both methodologies and inherent differences
in conditions, including temperature and averaging due to local sym-
metry. Thus, MD simulations naturally drive each individual ligand and
protein residue towards the nearest local energy minimum, resulting

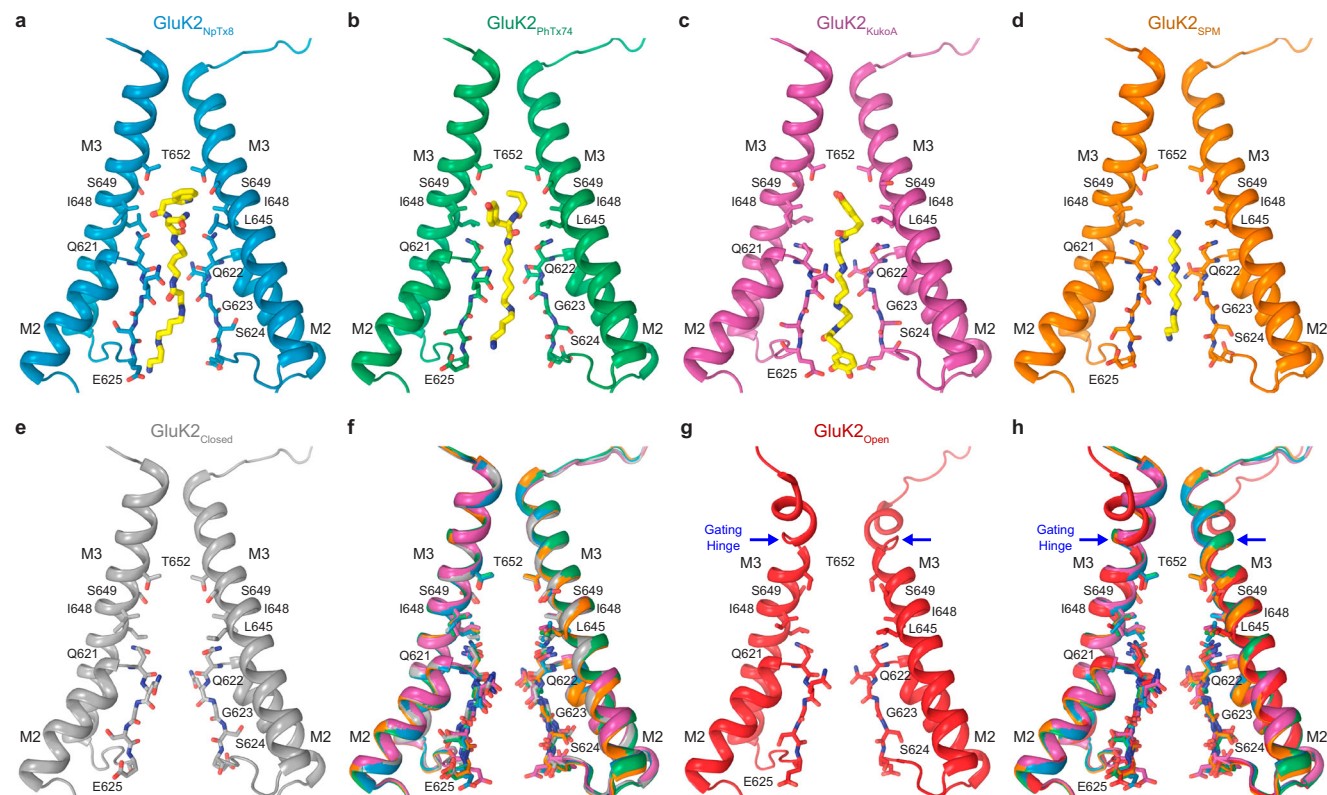

**Fig. 4 | Comparison of the blocker binding region in different structures.** Blocker binding region formed by M2 and M3 in GluK2$_{NpTx8}$ (**a**, blue), GluK2$_{PhTx74}$ (**b**, green), GluK2$_{KukoA}$ (**c**, violet), GluK2$_{SPM}$ (**d**, orange), GluK2$_{Closed}$ (**e**, gray, PDB ID: 8FWS), and GluK2$_{Open}$ (**g**, red, PDB ID: 9B35), with residues contributing to blocker binding shown in sticks. Only two (B and D) of four subunits are shown, with the front and back (A and C) subunits removed for clarity. The gating hinges are indicated by blue arrows in the GluK2$_{Open}$ structure. **f**, **h** Superposition of the blocker-bound structures from **a**–**d** with GluK2$_{Closed}$ (**f**) or GluK2$_{Open}$ (**h**).

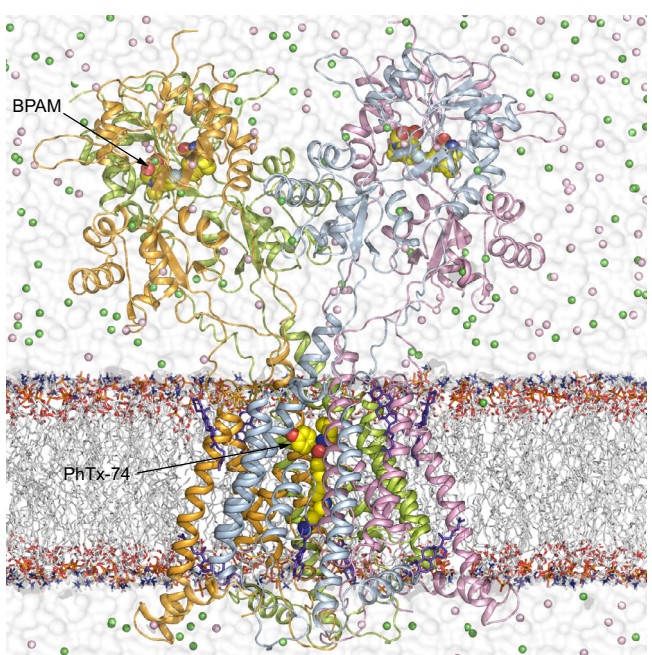

**Fig. 5 | MD simulation system.** A representation of the MD simulated GluK2$_{PhTx74}$ system with the receptor subunits A (green), B (pink), C (blue) and D (orange) shown as ribbons, lipid bilayer acyl chains in white and hydrophilic head groups as sticks, BPAM and PhTx-74 in yellow space-filling models, cholesterol as violet sticks, water as white continuum, and K$^+$ and Cl$^-$ ions as green and pink spheres, respectively.

in breaking the local symmetry due to entropic preferences, while the local symmetry constraints in cryo-EM structures may be pulling the system out of the local energy minima.

KukoA exhibited two semi-stable poses during simulations (Fig. 6a). The first pose remained almost the same as in the corresponding cryo-EM structure, where KukoA formed hydrogen bonds with the selectivity filter, similar to SPM. In this pose, the KukoA headgroup formed stable hydrophobic contacts with I648 across the central cavity. Water was mostly excluded from this region of the pore (Pose 1 in Fig. 6b). The second pose (Pose 2 in Fig. 6b) was characterized by shifting the blocker molecule extracellularly towards the pore center and letting both headgroups form hydrogen bonds with the protein in a more asymmetric manner. Also, a more compact conformation is generally more favorable than an extended conformation. In our simulations, both poses existed for similar periods of time. However, more extended simulations are needed to determine which one of the two poses is more energetically stable. Overall, the enhanced dynamic behavior of KukoA at the pore binding site and fewer interactions with the pore-lining residues correlate well with its lower potency compared to NpTx-8 and PhTx-74.

The least potent blocker SPM maintained the same pose throughout the entire MD simulation, which was very similar to its initial position in the corresponding cryo-EM structure (Fig. 6a). SPM formed stable hydrogen bonds with Q621 and Q622, as well as the side chains of E625 at the intracellular channel entrance. However, hydrogen bonds formed with surface protein groups solvated in water are typically weak, contributing less than 1 kcal/mol to the binding energy. In addition, the absence of a bulky head group in the SPM molecule allowed numerous water molecules to occupy the central cavity (Fig. 6b). Availability of water molecules to readily solvate polar atoms

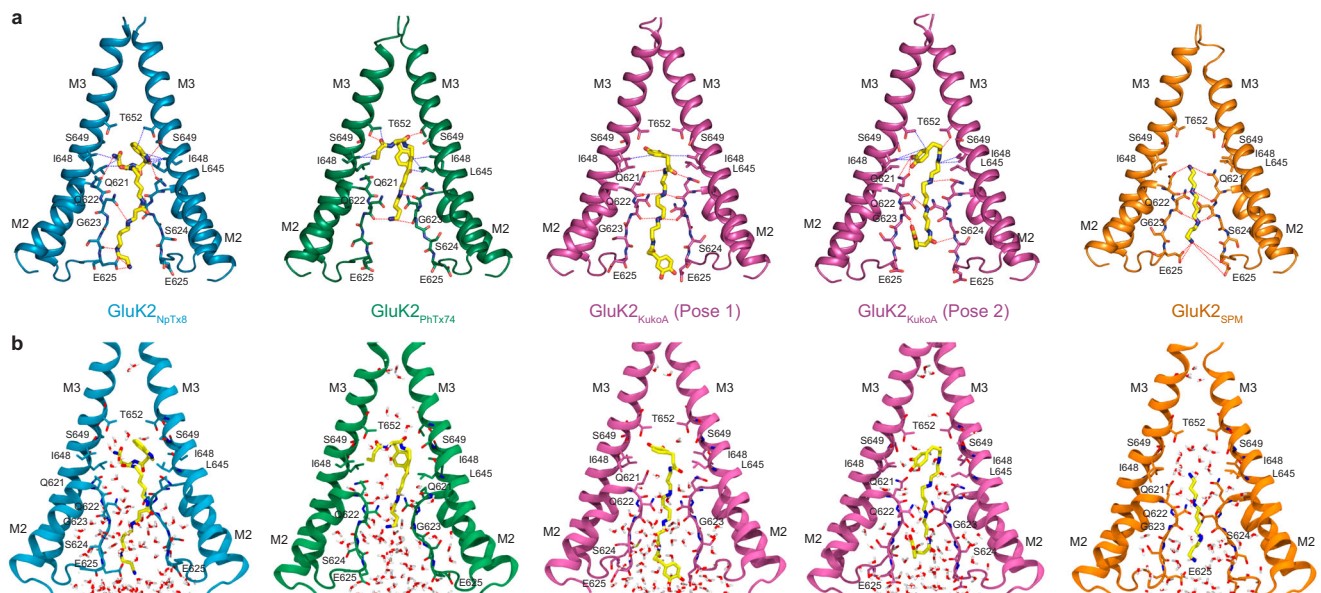

**Fig. 6 | Representative structures and water occupancy from MD simulations.**
**a** Close-up view of the blocker binding sites after the MD simulations of GluK2$_{NpTx8}$ (blue), GluK2$_{PhTx74}$ (green), GluK2$_{KukoA}$ (violet), and GluK2$_{SPM}$ (orange), with blocker molecules (yellow) and residues involved in their binding shown in stick representation. The red dashed lines show the hydrogen bonds between the blockers and the binding site residues, the blue dashed lines show the hydrophobic contacts between the blockers and the hydrophobic side chains of the protein residues. Only two (A and C) of four subunits are shown, with the front and back subunits (B and D) removed for clarity. For GluK2$_{KukoA}$, two representative structures (Pose 1 and Pose 2) from different parts of simulation are shown. **b** Similar views as in **a**, but with water molecules shown as red and white sticks.

of SPM suggests a low free energy barrier for blocker unbinding. Once the channel is in the open conformation, the corresponding fast dissociation of the blocker is manifested by the appearance of pronounced hook currents (Fig. 1b).

## Discussion

In this study, we investigated the trapping of four channel blockers of GluK2 KARs, NpTx-8, PhTx-74, KukoA, and SPM, which demonstrate a broad range of inhibitory potencies (Fig. 1). While all blocker molecules formed hydrogen bonds with the Q621 and Q622 residues (either with their backbone or side-chain oxygens), the more potent blockers NpTx-8 and PhTx-74 appear to do so in an asymmetric fashion, which may contribute to their stronger binding to the pore via increased entropy of their asymmetric interactions with identical 4-fold degenerate hydrogen bond acceptor sites. The weaker binder SPM retains its symmetric starting pose. In contrast, the second weakest binder, KukoA, adopted an alternative pose that retained the symmetry of the persistent hydrogen bonds of the tail.

Despite the four blockers displaying varying potencies and kinetics of KAR inhibition, they all interact with the channel gating machinery according to a trapping mechanism. The molecular basis of the trapping mechanism becomes apparent when comparing the structures of KAR channel in the closed blocked and open states[86,94] (Fig. 7). Similar to toxins and toxin-like molecules in AMPARs[67], the KAR channel blockers appear to fit the channel pore as keys fit a lock, with their positively-charged polyamine tails filling up the narrow, negatively-charged selectivity filter and bulky heads occupying the wider central cavity. In comparison to the open-state structure, it is easy to imagine how each one of these molecules can reach their binding site by entering the pore from extracellular space, with their positively charged polyamine tails moving forward toward the intracellular space, driven by the negative membrane voltage (Fig. 7). It is also clear that the negatively charged and narrow selectivity filter creates a barrier for the neutral bulky heads of NpTx-8, PhTx-74, and KukoA to traverse entirely through the pore into the intracellular space. Upon closure of the extracellular gate at the bundle crossing of

M3 helices, all four blockers also lose the ability to exit back to the extracellular space, thus being trapped inside the ion channel pore (Fig. 7).

Interestingly, all four blocker-bound structures, GluK2$_{NpTx8}$, GluK2$_{PhTx74}$, GluK2$_{KukoA}$, and GluK2$_{SPM}$, were solved in the absence of an agonist, which would typically open the channel and allow blockers to enter the pore. Therefore, these blockers either entered the pore of a ligand-free receptor or during infrequent pore openings due to contamination of solution with traces of agonist (glutamate). However, the possibility of glutamate contamination is unlikely; the structures obtained in this study are from multiple and independent protein purifications. In addition, we did not see an agonist-bound conformation of the receptor during extensive data processing. Further, binding or unbinding of iGluR blockers in the absence of an agonist were previously reported for both NMDARs[95] and AMPARs[96] and can be explained by dynamic properties of the protein with a plastic selectivity filter that allows spontaneous or blocker-induced openings of the gate. It may be possible to better understand how exactly blockers enter the channel without an agonist by running long MD simulations of the blocker-bound structures and observing putative ways for the blocker to escape the channel.

Calcium-permeable KARs are major contributors to neuropathologies[8,30,31]. Previous studies have elucidated the structural basis of competitive and noncompetitive antagonism of KARs[86,88]. However, the structural basis of how polyamines and polyamine-containing channel blockers inhibit KARs has remained poorly understood. Our study uncovers the structural mechanism of KAR channel block and provides molecular templates for the synthesis of new drug molecules targeting KARs. The broad spectrum of KAR channel blocker potencies demonstrated by NpTx-8, PhTx-74, KukoA, and SPM provides a reference for tuning this parameter. Finding the optimal potency, not necessarily the highest, is critical for drug design. For example, the only FDA-approved drugs targeting iGluRs, the ion channel blocker memantine acting on NMDARs and noncompetitive inhibitor perampanel acting on AMPARs, have rather intermediate potencies[76,97,98]. Nevertheless, new approaches of peptide-mediated

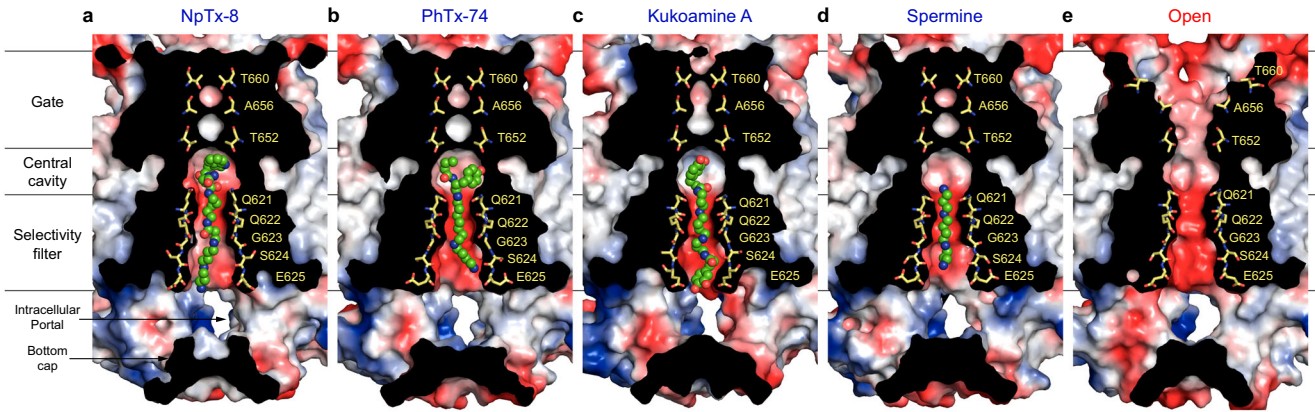

**Fig. 7 | Trapping of blockers in the closed pore of GluK2 receptor.** Frontal section of the TMD surface in the GluK2$_{PhTx74}$ (**a**), GluK2$_{NpTx8}$ (**b**), GluK2$_{KukoA}$ (**c**), GluK2$_{SPM}$ (**d**) and GluK2$_{Open}$ (**e**, PDB ID: 9B35) structures, with the blocker molecules shown as space-filling models (green) and residues contributing to the selectivity filter and gate shown in sticks (yellow). The surface is colored by electrostatic potential, blue being positively charged, red negatively charged, and white neutral.

targeting to achieve cell-specific modulation of iGluRs can make use of high-affinity drugs as well, as it was recently demonstrated by targeting high-affinity NMDAR channel blocker MK-801 to GLP-1 receptor-expressing brain regions for safe and effective obesity treatment[99].

## Methods

### Constructs
DNA for the full-length rat GluK2 (rGluK2, GenBank CAA77778.1), with V at position 567, C at position 571 and Q at position 621 (the Q/R site), was cloned into pEG BacMam vector for baculovirus-based protein expression in mammalian cells[100], with the C-terminal thrombin cleavage site (LVPRG), followed by eGFP and streptavidin affinity tag (WSHPQFEK).

### Protein expression and purification
The rGluK2 bacmid and baculovirus were made using standard methods[100]. The P1 and P2 viruses were produced in Sf9 cells (GIBCO, 12659017) and added to HEK 293S GnTI⁻ cells (ATCC, CRL-3022) incubated at 37 °C and 5% CO$_2$. 12–15 h post-transduction, cells were supplemented with 10 mM sodium butyrate, and the temperature was changed to 32 °C. Cells were harvested 72 h post-transduction using low-speed centrifugation (5500 g, 10 min), washed using 1X PBS pH 8.0, and centrifuged again (5500 g, 15 min). The cell pellet was resuspended in 50 ml of ice-cold lysis buffer consisting of 150 mM NaCl, 20 mM Tris pH 8.0, 5 mM βME, 0.8 µM aprotinin, 2 µg/ml leupeptin, 2 µM pepstatin A, 1 mM PMSF and 20 µM DNQX. Cells were lysed by three cycles of sonication using a Misonix sonicator with an amplitude of 8 for 3 min total, 15 s on and 15 s off. The cell-free lysate was centrifuged (9900 g, 15 min) to remove cell debris. To isolate cell membranes, the supernatant was subjected to ultracentrifugation (186,000 g, 1 h). Cell membranes were mechanically homogenized and solubilized for 2 h in a buffer containing 150 mM NaCl, 20 mM Tris-HCl pH 8.0, 5 mM βME, 20 µM DNQX, and 1% digitonin (Cayman Chemical Company, 14952). Insoluble material was removed by ultracentrifugation (186,000 g, 1 h). The supernatant was added to the pre-equilibrated streptavidin-linked resin (2 ml resin per 1 L of the initial cell culture). The mixture was rotated for 10–14 h at 4 °C. The protein-bound resin was washed with 25 ml of buffer containing 150 mM NaCl, 20 mM Tris-HCl pH 8.0, 0.05% digitonin, and 5 mM βME, and the protein was eluted in 12–15 mL of the same buffer supplemented with 2.5 mM D-desthiobiotin. To remove eGFP and the streptavadin affinity tag, the eluted protein was concentrated and subjected to thrombin digestion (1:200 w/w) at 22 °C for 90 min. The digest reaction was injected into a Superose 6 10/300 GL size-exclusion column (GE Healthcare) equilibrated with a buffer containing 150 mM NaCl, 20 mM Tris-HCl pH 8.0, 0.05% digitonin, and 5 mM βME. The peak fractions corresponding to tetrameric GluK2 were pooled, concentrated to ~5–6 mg/ml, and used for cryo-EM sample preparation. All steps, unless otherwise noted, were performed at 4 °C.

### Cryo-EM sample preparation and data collection
Purified rGluK2 at 5–6 mg/ml was incubated with channel blockers, 100 µM NpTx-8, 100 µM PhTX-74, 500 µM KukoA, or 1 mM SPM, for 12–15 h at 4 °C. For 20–30 min before grid preparation, the protein-blocker solutions were supplemented with 500 µM BPAM344. UltrAuFoil R 0.6/1.0, 300 mesh gold grids (EMS, Morrisville, NC) were used for cryo-EM sample preparation. Before sample application, the grids were treated in a PELCO easyGlow cleaning system (Ted Pella, 25 s, 15 mA) to make their surface hydrophilic. Subsequently, 3 µl of the protein sample was applied to each cryo-EM grid. Grids were made using a Vitrobot Mark IV (Thermo Fisher Scientific) set at 100% humidity and 4 °C, blot time of 3 s, blot force of 3, and a wait time of 30 s. The grids were imaged using Leginon 3.5 on a Titan Krios transmission electron microscope (Thermo Fisher Scientific) operating at 300 kV equipped with a post-column GIF Quantum energy filter with slit width set to 20 eV and equipped with either Gatan K3 (Gatan) or Falcon4 (Thermo Fisher Scientific) direct electron detection camera. The images were collected in counting or super-resolution mode across a defocus range of −1.0 to −2.0 µm.

### Cryo-EM data processing
Data were processed using cryoSPARC v4.4.1[101]. Movie frames were aligned using the Patch Motion Correction algorithm. Contrast transfer function (CTF) estimation was performed using the patch CTF estimation. Following CTF estimation, micrographs were manually inspected, and those with outliers in defocus values, ice thickness, and astigmatism, as well as micrographs with lower, predicted CTF-correlated resolution, were excluded from further processing (individually assessed for each parameter relative to the overall distribution). Particles were first picked using a blob picker and then by Topaz[102]. Junk particles were removed through successive rounds of two-dimensional classification. A set of particles corresponding to tetrameric rGluK2 was used for ab initio reconstruction. Subsequently, particles were cleaned up by successive rounds of heterogeneous refinement. During the processing of all four data sets, we noted that the transmembrane domain (TMD) and ligand-binding domain (LBD) regions of GluK2 displayed excellent density compared to the amino-terminal domain (ATD) due to flexibility in the ATD-LBD linker region

relative to the LBD-TMD region, consistent with the previous observations[86]. Accordingly, the particles were signal subtracted for the ATD and detergent micelle and cleaned up by multiple rounds of ab initio reconstruction and heterogeneous refinement. Finally, homogeneous, non-uniform, and local refinement with a focused mask around the LBD-TMD regions considerably improved the density of the LBD-TMD regions. The unsharpened maps obtained from cryoSPARC were post-processed in Phenix through anisotropic sharpening, significantly improving the density of the LBD-TMD linker regions. Structural biology applications employed in this project adhered to and were configured by SBGrid. Data processing details are summarized in Table 1.

## Model building and refinement

The models of blocker-bound GluK2, GluK2$_{NpTx8}$, GluK2$_{PhTx74}$, GluK2$_{KukoA}$, and GluK2$_{SPM}$ were built in Coot[103] using the corresponding experimental cryo-EM densities and the GluK2$_{Closed}$ structure (8FWS)[86] as a guide. The models were tested for overfitting by shifting their coordinates by 0.5 Å (using shake) in Phenix 1.18[104], refining each shaken model against a corresponding unfiltered half map, and generating densities from the resulting models in Chimera. The resulting models were real space refined in Phenix 1.18 and visualized in Chimera[105] or Pymol 2.5.2[106].

## Patch-clamp recordings

DNA encoding rGluK2 (described in the Construct section) was introduced into a pIRES plasmid for expression in eukaryotic cells engineered to produce green fluorescent protein via a downstream internal ribosome entry site[107]. HEK 293 cells (ATCC, Cat#CRL-1573) grown on glass coverslips in 35 mm dishes were transiently transfected with 1–5 μg of plasmid DNA using Lipofectamine 2000 Reagent (Invitrogen). Recordings were made 24–48 h after transfection at room temperature. Currents from whole cells, typically held at a −60 mV potential, were recorded using Axopatch 200B amplifier (Molecular Devices, LLC), filtered at 5 kHz, and digitized at 10 kHz using low-noise data acquisition system Digidata 1440A and pCLAMP 10.2 software (Molecular Devices, LLC). The external solution contained: 150 mM NaCl, 2.4 mM KCl, 4 mM CaCl₂, 4 mM MgCl₂, 10 mM HEPES pH 7.3. 7 mM NaCl was added to the extracellular activating solution containing 3 mM Glu to improve visualization of the border between two solutions coming out of a two-barrel theta glass pipette, which allowed its more precise positional adjustment for faster solution exchange. The internal solution contained: 150 mM CsF, 10 mM NaCl, 10 mM EGTA, 20 mM HEPES pH 7.3. Before the experiment, cells were treated in extracellular solution with 0.3 mg/ml ConA (Sigma) for 5–15 min and then transferred to solution without ConA. Rapid solution exchange was achieved with a two-barrel theta glass pipette controlled by a piezoelectric translator. Typical 10–90% rise times were 200–300 μs, as measured from junction potentials at the open tip of the patch pipette after recordings. Data analysis was performed using Origin 2023 software (OriginLab Corporation).

## System preparation for molecular dynamics simulations

Initial atomic coordinates for Molecular Dynamics (MD) simulations were obtained from the GluK2$_{NpTx8}$, GluK2$_{PhTx74}$, GluK2$_{KukoA}$, and GluK2$_{SPM}$ structures. Each structure includes LBD and TMD of the full-length receptor, starting with the N-terminal residue S429 of the LBD and ending with the C-terminal residue R874 of the TMD. For all the simulation systems, protein, blocker molecules (NpTx-8, PhTx-74, KukoA, and SPM), BPAM, cholesterol, and experimental lipids from the cryo-EM were kept, and all the other molecules were removed. The GluK2$_{NpTx8}$ system did not include cholesterol molecules as it lacks them in cryo-EM density. Each simulation box was constructed in CHARMM-GUI membrane builder[108,109] by inserting the protein into a POPC bilayer and solvating it with TIP3P water molecules and 150 mM

KCl. The systems were set up for MD simulations using the "tleap" module of the AmberTools20 package[110]. Parametrization of all the ligands was carried out using the general AMBER force field (GAFF)[111]. The total number of atoms in the final simulation boxes was 311,702 for GluK2$_{NpTx8}$, 313,114 for GluK2$_{PhTx74}$, 312,549 for GluK2$_{KukoA}$, and 315,207 for GluK2$_{SPM}$ system. Each simulation system comprised approximately 73,000 water molecules, 200 K⁺ and Cl⁻ ions, and 500 lipid molecules.

## Molecular dynamics simulation protocols

"pmemd.cuda" program of the Amber20 molecular dynamics software package was used for all MD simulations[110]. Amber FF99SB−ILDN[112] force field parameters were used for protein and ions, TIP3P model for water, and Lipid14[113] force field parameters for lipids. All equilibration and production simulations were performed in an NPT ensemble at 300 K temperature and 1 bar pressure with anisotropic pressure scaling. The Langevin thermostat with a collision frequency of 1 ps⁻¹ was used to control the temperature, and the pressure was maintained using a Berendsen barostat with a relaxation time of 1 ps, as implemented in Amber20. All covalent bonds involving hydrogen atoms were constrained using the SHAKE algorithm[114], with the integration time step of 2 fs. The long-range electrostatic interaction calculations were performed using the Particle Mesh Ewald (PME) method[115], with a non-bonded interaction cutoff radius of 10 Å. Periodic boundary conditions were applied in all directions.

Each system was minimized prior to MD simulations while keeping restraints on protein Cα and ligand (BPAM and blocker) heavy atoms. Next, water and ions were equilibrated at constant volume MD simulations as the temperature was gradually increased from 0 to 300 K, with all protein, ligand, and lipid heavy atoms harmonically restrained at their energy-minimized positions with the force constant of 40 kcal mol⁻¹ Å⁻². The systems were then equilibrated for 100 ns at constant pressure MD simulations, gradually releasing the restraints on the protein and ligands to 0.5 kcal mol⁻¹ Å⁻². Production simulations (one run per blocker) were carried out for ~600 ns for GluK2$_{NpTx8}$, GluK2$_{PhTx74}$, GluK2$_{KukoA}$, and ~800 ns for GluK2$_{SPM}$ systems without restraints.

## Molecular dynamics trajectory analysis

Post-processing and analysis of the trajectories were carried out using CPPTRAJ[116] module of AmberTools20 and VMD 1.9.4[117]. VMD 1.9.4 was used to visualize trajectories, and PyMOL[106] (The PyMOL Molecular Graphics System, Version 2.0 Schrödinger, LLC.) was used to generate simulation snapshot figures. Heavy-atom contact frequencies were calculated in VMD when any heavy-atom of the protein and heavy-atom of the blocker molecule is within 4 Å. Hydrogen bond and hydrophobic interaction analysis were carried out in CPPTRAJ. To determine the list of potential hydrogen bonds, a geometric criterion where the donor-acceptor distance is within 3.6 Å with the donor-hydrogen-acceptor angle cutoff of 90° was used. Hydrophobic contacts were determined when any two pairs of carbon atoms or any carbon and any sulfur atom were at a distance shorter than 4 Å. The persistence of these contacts was reported with duty fraction as a percentage which is defined as the ratio of the duration of bond on time and the total simulation time. Protein-ligand contacts analysis was performed in VMD. Ligand RMSD was computed for heavy atoms of the ligand with reference to their initial positions, with each frame of the trajectory aligned using the coordinates of protein Cα atoms for residues 631 to 661 in the M3 helix. Protein TMD RMSD was computed for Cα atoms of the M1, M2, M3, and M4 TMD helices. Representative structures were generated by averaging the coordinates over the production simulation and selecting the lowest RMSD structure. For GluK2$_{SPM}$ and GluK2$_{NpTx8}$ systems, representative structures were calculated from the entire production trajectory. For GluK2$_{KukoA}$ system, the first representative structure (GluK2$_{KukoA}$-1) was extracted

from the first 250 ns of the production trajectory, and the second representative structure (GluK2$_{KukoA}$-2) was extracted from the 350–500 ns portion of the trajectory in which the blocker molecule adopted a new conformation and stabilized. For GluK2$_{PhTx74}$ system, the representative structure was extracted from the production trajectory after the blocker conformation is stabilized by excluding the first 120 ns.

**Statistics and reproducibility.** Statistical analysis (Figs. 1a–c and 4a–b) was performed using Origin 2023. Statistical significance was calculated using One-Way ANOVA followed by Fisher's least significant difference test. In all figure legends, $n$ represents the number of independent biological replicates. All quantitative data were presented as mean ± SEM.

### Reporting summary
Further information on research design is available in the Nature Portfolio Reporting Summary linked to this article.

## Data availability
The data that support this study are available from the corresponding author upon request. Cryo-EM density maps have been deposited to the Electron Microscopy Database (EMDB) under the accession codes EMD-47295 for GluK2$_{PhTx74}$, EMD-47296 for GluK2$_{NpTx8}$, EMD-47297 for GluK2$_{SPM}$, and EMD-47298 for GluK2$_{KukoA}$. The corresponding atomic coordinates have been deposited to the Protein Data Bank (PDB) under the accession codes 9DXQ for GluK2$_{PhTx74}$, 9DXR for GluK2$_{NpTx8}$, 9DXS for GluK2$_{SPM}$, and 9DXT for GluK2$_{KukoA}$. The accession codes for previously published structures that were used for model building or illustrations: 8FWS for GluK2$_{closed}$, and 9B35 for GluK2$_{open}$. The molecular dynamics simulation data without lipid and water coordinates, together with topology files, have been deposited in Zenodo [10.5281/zenodo.14009849]. The repository also includes the full initial coordinates (with waters and lipids) for the production runs as well as the representative structures that correspond to the structures in Fig. 6, as described in the Methods section. The raw simulation data have not been deposited due to their large size, but access can be obtained by contacting the authors. The source data underlying Fig. 1c, e are provided as a Source Data file. Source data are provided with this paper.

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

## Acknowledgements

We thank Robert Grassucci and Zhening Zhang (Columbia University Cryo-Electron Microscopy Center), Gabriella Angiulli (New York Structural Biology Center/ National Center for CryoEM Access and Training), Patrick Mitchell, Ian Fries (Stanford Linear Accelerator Center/National Accelerator Laboratory) and Sean Mulligan (Pacific Northwest cryo-EM Center (PNCC) for help with microscope operation and data collection. Some of this work was performed at the Columbia University Cryo-Electron Microscopy Center. A portion of this research was supported by NIH grant U24GM129547 and performed at the PNCC at OHSU and accessed through EMSL (grid.436923.9), a DOE Office of Science User Facility sponsored by the Office of Biological and Environmental Research. Some of this work was performed at the National Center for CryoEM Access and Training (NCCAT) and the Simons Electron Microscopy Center located at the New York Structural Biology Center, supported by the NIH Common Fund Transformative High Resolution Cryo-Electron Microscopy program (U24 GM129539) and by grants from the Simons Foundation (SF349247) and NY State Assembly Majority. Some of this work was performed at the Stanford-SLAC Cryo-EM Center (S2C2), supported by the National Institutes of Health Common Fund Transformative High Resolution Cryo-Electron Microscopy program (U24 GM129541). S.P.G. was supported by the NIH (NS139087). A.I.S. was supported by the NIH (NS083660, NS107253, AR078814, CA206573). M.G.K. was supported by the NSF MCB-1818213 and grant of time XSEDE NSF MCB180173.

## Author contributions

S.P.G., M.V.Y., and A.I.S. conceptualized the project and designed the experiments. S.P.G. and M.V.Y. made constructs for protein expression and electrophysiology. S.P.G. and L.Y.Y. performed protein purification, made cryo-EM grids, and collected cryo-EM data. S.P.G., L.Y.Y., and T.P.N. processed cryo-EM data. M.V.Y. performed patch-clamp recordings and electrophysiological data analysis. M.A. and M.G.K. designed and performed MD simulations and analysis. S.P.G. and A.I.S. built molecular models. K.S. synthesized NpTx-8. S.P.G., M.V.Y., M.A., L.Y.Y., T.P.N., M.G.K., and A.I.S. wrote the manuscript. A.I.S. supervised the project.

## Competing interests

The authors declare no competing interests.
