## [Transparent Peer Review file · Nature Communications]

Trapping of spermine, Kukoamine A, and polyamine toxin blockers in GluK2 kainate receptor channels

Corresponding Author: Dr Alexander Sobolevsky

Version 0:

Reviewer comments:

Reviewer #1

(Remarks to the Author)

This manuscript provides a comprehensive examination of the inhibitory mechanisms of kainate receptors (KARs), which play a crucial role in synaptic transmission and are implicated in various neurological disorders. While previous studies have underscored the importance of KARs in these processes, the precise molecular details of how ion channel blockers interact with KARs have remained elusive until now. This study fills a critical gap by offering high-resolution structural data on blocker interactions within the GluK2 subunit, something not previously documented in such detail. The research focuses on four blockers - NpTx-8, PhTx-74, KukoA, and spermine (SPM) - revealing how these blockers trap themselves within the ion channel pore of GluK2 and prevent ion flow. The potencies of these blockers vary due to their specific interactions with residues in the pore, highlighting the structural and dynamic features that drive their effectiveness.

By integrating cryo-electron microscopy (cryo-EM), electrophysiology, and molecular dynamics (MD), the authors offer a detailed understanding of how these ion channel blockers interact with KARs. The cryo-EM captures structural details, while MD illuminates the dynamic behaviors of the blockers in near-physiological conditions. This multidimensional approach, supplemented by functional assays, provides new insights into how variations in binding dynamics influence blocker potency and effectiveness. Importantly, the combination of methods provides robust support for the study's conclusions, with no additional evidence needed to validate the core claims.

The quality of the experiments is exceptionally high, with a comprehensive analysis that integrates structural insights, computational modeling, and functional validation. The methodology is fully described in detail, allowing for reproducibility, which meets the expected standards in the field. This study contributes valuable knowledge to neurobiology, particularly in understanding how KAR inhibition can be leveraged for therapeutic purposes in conditions such as epilepsy and neurodegenerative diseases. Compared to previous literature, this work stands out by filling a crucial gap in understanding the molecular-level interactions between blockers and KARs, moving the field forward with fresh key insights.

No major flaws in data analysis or interpretation are apparent, and the methodology is sound. These findings could catalyze future developments in therapeutic approaches targeting KAR dysfunction, enhancing our understanding of neuronal excitability and synaptic regulation, making this research impactful for both academic researchers and drug developers.

Reviewer #2

(Remarks to the Author)

In this study, Gangwar et al. explored the structural mechanism of inhibition of GluK2 Kainate receptor (KAR) channels by four channel blockers i.e. NpTx-8, PhTx-74, KukoA, and SPM. They showed that these channel blockers behaved like trapping blockers using electrophysiological experiments. Further, they determined cryoEM structures of GluK2 KAR homotetramers in complex with these blockers. All the structures were in closed state where blockers occupied the ion channel pore. These structures help to understand the molecular details of binding inside the channel pore and elucidates the reason for varying potencies of these blockers. Using molecular dynamics simulations, they explored the dynamics of interaction with the channel that highlighted the varying degrees of flexibility in binding poses of these blockers. This study is well executed and advances our understanding of the structural mechanism of KAR channel inhibition. The manuscript is suitable for publication in its current form. However, certain aspects need more clarification or modifications to benefit the

manuscript.

1. The authors should include figure showing the density of the channel blockers fitted with the corresponding structure models.
2. The blockers densities seem to have low resolution compared to the surrounding protein residues which may be result of heterogeneity. Doing further 3D classification focusing on only the TMD region or after symmetry-expansion may help further to improve the blocker density.
3. Line 332-333: The spontaneous opening of the gate to allow the blocker to enter the channel seems less probable given the lack of evidence to show that kainite receptor channels can sample open state in the absence of agonist. Is there a possibility that these channel blockers induce opening of the channel and when they move inside the pore, the channel closes trapping the blocker. If so, then in the presence of the blocker, the channel should transiently open and then get blocked. Also, why do authors think that there is possibility of blocker entry through side portals given the fact that these molecules are highly charged and cannot enter membrane bilayer.
4. MD simulation results suggest that one of the common key determinants of binding among all the four blockers is hydrogen bonding with Q621 and Q622. However, the way PhTx-74 is modeled in the cryoEM density, this interaction seems to be absent. Further, the polar residues in the pore seem to surround the hydrophobic region of the blocker which is thermodynamically not very favorable. If the cryoEM structure represents some high energy state (unlike the seemingly low-energy state obtained in the MD simulation) then why this high energy state is abundant in the cryoEM sample?

Version 1:

Reviewer comments:

Reviewer #2

(Remarks to the Author)

The authors have satisfactorily addressed all the concerns.

We thank all Reviewers for their excellent suggestions that have led to significant improvement of this manuscript. We have made changes throughout the manuscript with the details outlined in our responses below.

Reviewer #1 (Remarks to the Author):

This manuscript provides a comprehensive examination of the inhibitory mechanisms of kainate receptors (KARs), which play a crucial role in synaptic transmission and are implicated in various neurological disorders. While previous studies have underscored the importance of KARs in these processes, the precise molecular details of how ion channel blockers interact with KARs have remained elusive until now. This study fills a critical gap by offering high-resolution structural data on blocker interactions within the GluK2 subunit, something not previously documented in such detail. The research focuses on four blockers - NpTx-8, PhTx-74, KukoA, and spermine (SPM) - revealing how these blockers trap themselves within the ion channel pore of GluK2 and prevent ion flow. The potencies of these blockers vary due to their specific interactions with residues in the pore, highlighting the structural and dynamic features that drive their effectiveness.

By integrating cryo-electron microscopy (cryo-EM), electrophysiology, and molecular dynamics (MD), the authors offer a detailed understanding of how these ion channel blockers interact with KARs. The cryo-EM captures structural details, while MD illuminates the dynamic behaviors of the blockers in near-physiological conditions. This multidimensional approach, supplemented by functional assays, provides new insights into how variations in binding dynamics influence blocker potency and effectiveness. Importantly, the combination of methods provides robust support for the study's conclusions, with no additional evidence needed to validate the core claims.

The quality of the experiments is exceptionally high, with a comprehensive analysis that integrates structural insights, computational modeling, and functional validation. The methodology is fully described in detail, allowing for reproducibility, which meets the expected standards in the field. This study contributes valuable knowledge to neurobiology, particularly in understanding how KAR inhibition can be leveraged for therapeutic purposes in conditions such as epilepsy and neurodegenerative diseases. Compared to previous literature, this work stands out by filling a crucial gap in understanding the molecular-level interactions between blockers and KARs, moving the field forward with fresh key insights.

No major flaws in data analysis or interpretation are apparent, and the methodology is sound. These findings could catalyze future developments in therapeutic approaches targeting KAR dysfunction, enhancing our understanding of neuronal excitability and synaptic regulation, making this research impactful for both academic researchers and drug developers.

We thank Reviewer #1 for the positive and generous assessment of our work.

Reviewer #2 (Remarks to the Author):

In this study, Gangwar et al. explored the structural mechanism of inhibition of GluK2 Kainate receptor (KAR) channels by four channel blockers i.e. NpTx-8, PhTx-74, KukoA, and SPM. They

showed that these channel blockers behaved like trapping blockers using electrophysiological experiments. Further, they determined cryoEM structures of GluK2 KAR homotetramers in complex with these blockers. All the structures were in closed state where blockers occupied the ion channel pore. These structures help to understand the molecular details of binding inside the channel pore and elucidates the reason for varying potencies of these blockers. Using molecular dynamics simulations, they explored the dynamics of interaction with the channel that highlighted the varying degrees of flexibility in binding poses of these blockers. This study is well executed and advances our understanding of the structural mechanism of KAR channel inhibition. The manuscript is suitable for publication in its current form.

We thank Reviewer #2 for the kind assessment of our work.

However, certain aspects need more clarification or modifications to benefit the manuscript.

1. The authors should include figure showing the density of the channel blockers fitted with the corresponding structure models.

New panels d-f showing density for the blockers fitted with the corresponding models has now been added to Extended Data Figure 5.

2. The blockers densities seem to have low resolution compared to the surrounding protein residues which may be result of heterogeneity. Doing further 3D classification focusing on only the TMD region or after symmetry-expansion may help further to improve the blocker density.

We absolutely agree with Reviewer #2 that weaker density for the blockers compared to the surrounding protein is the result of heterogeneity, which is due to dynamic behavior of the blocker at the binding site (see MD simulations) as well as different poses that the blocker can adopt at this pore location due to channel symmetry. The protein in this location has ~4-fold rotational symmetry and the blocker molecules that have C1 symmetry can be fitted here in 4 different but equivalent ways (see new panels d-f in Extended Data Figure 5). We have extensively tried symmetry expansion and focused classifications/refinements during our data processing but have not succeeded in making substantial improvements. We think that the major difficulties in successfully applying these techniques in our case are (1) small size of the blocker molecules and (2) their location right at the axis of ~4-fold rotational symmetry (where noise accumulates dramatically). The corresponding discussion has been added to the text (lines 184-193).

3. Line 332-333: The spontaneous opening of the gate to allow the blocker to enter the channel seems less probable given the lack of evidence to show that kainite receptor channels can sample open state in the absence of agonist. Is there a possibility that these channel blockers induce opening of the channel and when they move inside the pore, the channel closes trapping the blocker. If so, then in the presence of the blocker, the channel should transiently open and then get blocked. Also, why do authors think that there is possibility of blocker entry through side

portals given the fact that these molecules are highly charged and cannot enter membrane bilayer.

We agree with Reviewer #2 that the possibility of these highly charged blocker molecules entering the pore through side portals is low. We have removed this suggestion from the text. We also thank Reviewer #2 for the suggestion about the blocker-induced entry, which we have now added to the text (lines 346-349). While it is difficult to think about experimental evidence of such blocker-induced entry, there is evidence of blocker-induced escape from the trapped state in closely related AMPA receptors (ref. 96). Given the reversibility of kinetic transitions in ion channels, these data support the feasibility of the reverse process of blocker-induced entry. We have cited the corresponding paper in the text (line 348).

4. MD simulation results suggest that one of the common key determinants of binding among all the four blockers is hydrogen bonding with Q621 and Q622. However, the way PhTx-74 is modeled in the cryoEM density, this interaction seems to be absent. Further, the polar residues in the pore seem to surround the hydrophobic region of the blocker which is thermodynamically not very favorable. If the cryoEM structure represents some high energy state (unlike the seemingly low-energy state obtained in the MD simulation) then why this high energy state is abundant in the cryoEM sample?

This is a very interesting question raised by Reviewer #2. Blocker molecule models were fitted into cryo-EM densities the best possible way in Coot and then refined in Phenix. In all cases except PhTx-74, the orientations of molecules in cryo-EM models are very similar to the poses predicted by MD simulations (with Q621 and Q622 making hydrogen bonding with the blocker molecules). The PhTx-74 case is certainly important and we present it as an example of a discrepancy between cryo-EM data modeling and MD simulations (both have limitations and neither one could be considered as giving the ultimately true result). In addition to the obvious limitations of the methodologies, there are differences in the conditions of the experiment and simulations, such as different temperatures and different averaging due to the system's symmetry. The MD simulations will naturally drive each individual ligand and protein residue in the system towards the nearest local energy minimum, which will result in breaking the symmetry locally due to the entropic preferences; while in the experimental structures the symmetry of the system is preserved and may result in the system being drawn out of the energy local minima for individual groups. Despite this disparity, both cryo-EM and MD models support our structural mechanism of trapping. Therefore, we have decided not to attempt changing the fitting into cryo-EM models (which served as the starting points for the MD simulations) but rather discuss the differences between the cryo-EM and MD data in a discussion that we have added to the text of the manuscript (lines 278-287).